# RelWalk – A Latent Variable Model Approach to Knowledge Graph Embedding

## Abstract

Knowledge Graph Embedding (KGE) is the task of jointly learning entity and relation embeddings for a given knowledge graph. Existing methods for learning KGEs can be seen as a two-stage process where (a) entities and relations in the knowledge graph are represented using some linear algebraic structures (embeddings), and (b) a scoring function is defined that evaluates the strength of a relation that holds between two entities using the corresponding relation and entity embeddings. Unfortunately, prior proposals for the scoring functions in the first step have been heuristically motivated, and it is unclear as to how the scoring functions in KGEs relate to the generation process of the underlying knowledge graph. To address this issue, we propose a generative account of the KGE learning task. Specifically, given a knowledge graph represented by a set of relational triples $(h, R, t)$, where the semantic relation $R$ holds between the two entities $h$ (head) and $t$ (tail), we extend the random walk model (Arora et al., 2016a) of word embeddings to KGE. We derive a theoretical relationship between the joint probability $p(h, R, t)$ and the embeddings of $h$, $R$ and $t$. Moreover, we show that marginal loss minimisation, a popular objective used by much prior work in KGE, follows naturally from the log-likelihood ratio maximisation under the probabilities estimated from the KGEs according to our theoretical relationship. We propose a learning objective motivated by the theoretical analysis to learn KGEs from a given knowledge graph. The KGEs learnt by our proposed method obtain state-of-the-art performance on FB15K237 and WN18RR benchmark datasets, providing empirical evidence in support of the theory.

## 1 Introduction

Knowledge graphs such as Freebase (Bollacker et al., 2008) organise information in the form of graphs, where entities are represented by vertices in the graph and the relation between two entities is represented by the edge that connects the corresponding two vertices. By embedding entities and relations that exist in a knowledge graph in some (possibly lower-dimensional and latent) space we can infer previously unseen relations between entities, thereby expanding a given knowledge graph (Nickel et al., 2016; Yang et al., 2015; Lin et al., 2015; Nickel et al., 2011; Trouillon et al., 2016; Wang et al., 2017; Bordes et al., 2011).

Existing KGE methods can be seen as involving two main steps. First, given a knowledge graph represented by a set of relational triples $(h, R, t)$, where a semantic relation $R$ holds between a head entity $h$ and a tail entity $t$, entities and relations are represented using some mathematical structures such as vectors, matrices or tensors. Second, a scoring function is proposed that evaluates the *relational strength* of a triple $(h, R, t)$ and entity and relation embeddings that optimise the defined scoring function are learnt using some optimisation method. Table 1 shows some of the scoring functions proposed in prior work in KGE learning.

Despite the wide applications of entity and relation embeddings created via KGE methods, the existing scoring functions are motivated heuristically to capture some geometric requirements of the embedding space. For example, TransE (Bordes et al., 2011) assumes that the entity and relation embeddings co-exist in the same (possibly lower dimensional) vector space and translating (shifting) the head entity embedding by the relation embedding must make it closer to the tail entity embedding, whereas ComplEx (Trouillon et al., 2016) models the asymmetry in relations using

| Model | Score function $f(h, R, t)$ | Relation parameters |
|---|---|---|
| Unstructured (Bordes et al., 2011) | $\|\boldsymbol{h} - \boldsymbol{t}\|_{\ell_{1/2}}$ | none |
| Structured embeddings (Bordes et al., 2011) | $\|\mathbf{R}_1\boldsymbol{h} - \mathbf{R}_2\boldsymbol{t}\|_{\ell_{1,2}}$ | $\mathbf{R}_1, \mathbf{R}_2 \in \mathbb{R}^{d \times d}$ |
| TransE (Bordes et al., 2011) | $\|\boldsymbol{h} + \boldsymbol{R} - \boldsymbol{t}\|_{\ell_{1/2}}$ | $\boldsymbol{R} \in \mathbb{R}^d$ |
| DistMult (Yang et al., 2015) | $\langle \boldsymbol{h}, \boldsymbol{R}, \boldsymbol{t} \rangle$ | $\boldsymbol{R} \in \mathbb{R}^d$ |
| RESCAL (Nickel et al., 2011) | $\boldsymbol{h}^\top \mathbf{R} \boldsymbol{t}$ | $\mathbf{R}^{d \times d}$ |
| ComplEx (Trouillon et al., 2016) | $\langle \boldsymbol{h}, \boldsymbol{R}, \bar{\boldsymbol{t}} \rangle$ | $\boldsymbol{R} \in \mathbb{C}^d$ |

Table 1: Score functions proposed in selected prior work on KGE. Entity embeddings $\boldsymbol{h}, \boldsymbol{t} \in \mathbb{R}^d$ are vectors in all models, except in ComplEx where $\boldsymbol{h}, \boldsymbol{t} \in \mathbb{C}^d$. Here, $\boldsymbol{x}_{\ell_{1/2}}$ denotes either $\ell_1$ or $\ell_2$ norm of the vector $\boldsymbol{x}$. In ComplEx, $\bar{\boldsymbol{x}}$ is the elementwise complex conjugate, and $\langle \cdot, \cdot, \cdot \rangle$ denotes the component-wise multi-linear inner-product.

the component-wise multi-linear inner-product among entity and relation embeddings. Relational triples extracted from a given knowledge graph are used as positive training instances, whereas pseudo-negative (Bordes et al., 2011) instances are automatically generated by randomly corrupting positive instances. Finally, KGE are learnt such that the prediction loss computed over the positive and negative instances is minimised.

Despite the good empirical performances of the existing KGE methods, theoretical understanding of KGE methods is comparatively under developed. For example, it is not clear how the heuristically defined KGE objectives relate to the generative process of a knowledge graph. In this paper, we attempt to fill this void by providing a theoretical analysis of KGE. Specifically, in section 2, we propose a generative process where we explain the formation of a relation $R$ between two entities $h$ and $t$ using the corresponding relation and entity embeddings. Following this generative story, we derive a relationship between the probability of $R$ holding between $h$ and $t$, $p(h, t \mid R)$, and the embeddings of $R$, $h$ and $t$. Interestingly, the derived relationship is not covered by any of the previously proposed heuristically-motivated scoring functions, providing the first-ever KGE method with a provable generative explanation.

Next, in section 3, we show that the *margin loss*, which has been popularly used as a training objective in prior work on KGE, naturally arises as the log-likelihood ratio computed from $p(h, t \mid R)$. Based on this result, we derive a training objective that we subsequently optimise for learning KGEs that satisfy our theoretical relationship. Using standard benchmark datasets proposed in prior work on KGE learning, we evaluate the learnt KGEs on a link prediction task and a triple classification task. Experimental results show that the learnt KGEs obtain state-of-the-art performance on FB15K237 and WN18RR benchmarks, thereby providing empirical evidence to support the theoretical analysis.

## 2 RELATIONAL WALK

Let us consider a knowledge graph $\mathcal{D}$ where the *knowledge* is represented by relational triples $(h, R, t) \in \mathcal{D}$. Here, $R$ is a relational predicate of two arguments, where $h$ (*head*) and $t$ (*tail*) entities respectively filling the first and second arguments. We assume relations to be asymmetric in general. In other words, if $(h, R, t) \in \mathcal{D}$ then it does not necessarily follow that $(t, R, h) \in \mathcal{D}$. The goal of KGE is to learn embeddings (representations) for the relations and entities in the knowledge graph such that the entities that participate in similar relations are embedded closely to each other in the entity embedding space, while at the same time relations that hold between similar entities are embedded closely to each other in the relational embedding space. We call the learnt entity and relation embeddings collectively as KGEs. Following prior work on KGE (Bordes et al., 2011; Trouillon et al., 2016; Yang et al., 2015), we assume that entities and relations are embedded in the same vector space, allowing us to perform linear algebraic operations using the embeddings in the same vector space.

Let us consider a random walk characterised by a time-dependent *knowledge vector* $\boldsymbol{c}_k$, where $k$ is the current time step. The knowledge vector represents the knowledge we have about a particular group of entities and relations that express some facts about the world. For example, the knowledge that we have about people that are employed by companies can be expressed using entities of classes

such as people and organisation, using relations such as CEO-of, employed-at, works-for, etc. We assume that entities $h$ and $t$ are represented by time-independent $d$-dimensional vectors, respectively $\boldsymbol{h}, \boldsymbol{t} \in \mathbb{R}^d$.

We assume the task of generating a relational triple $(h, R, t)$ in a given knowledge graph to be a two-step process as described next. First, given the current knowledge vector at time $k$, $\boldsymbol{c} = \boldsymbol{c}_k$ and the relation $R$, we assume that the probability of an entity $h$ satisfying the first argument of $R$ to be given by (1).

$$p(h \mid R, \boldsymbol{c}) = \frac{1}{Z_c} \exp\left(\boldsymbol{h}^\top \mathbf{R}_1 \boldsymbol{c}\right). \tag{1}$$

Here, $\mathbf{R}_1 \in \mathbb{R}^{d \times d}$ is a relation-specific orthogonal matrix that evaluates the appropriateness of $h$ for the first argument of $R$. For example, if $R$ is the CEO-of relation, we would require a person as the first argument and a company as the second argument of $R$. However, note that the role of $\mathbf{R}_1$ extends beyond simply checking the types of the entities that can fill the first argument of a relation. For our example above, not all people are CEOs and $\mathbf{R}_1$ evaluates the likelihood of a person to be selected as the first argument of the CEO-of relation. $Z_c$ is a normalisation coefficient such that $\sum_{h \in \mathcal{V}} p(h \mid R, \boldsymbol{c}) = 1$, where the vocabulary $\mathcal{V}$ is the set of all entities in the knowledge graph.[1]

After generating $h$, the state of our random walker changes to $\boldsymbol{c}' = \boldsymbol{c}_{k+1}$, and we next generate the second argument of $R$ with the probability given by (2).

$$p(t \mid R, \boldsymbol{c}') = \frac{1}{Z_{c'}} \exp\left(\boldsymbol{t}^\top \mathbf{R}_2 \boldsymbol{c}'\right). \tag{2}$$

Here, $\mathbf{R}_2 \in \mathbb{R}^{d \times d}$ is a relation-specific orthogonal matrix that evaluates the appropriateness of $t$ as the second argument of $R$. $Z_{c'}$ is a normalisation coefficient such that $\sum_{t \in \mathcal{V}} p(t \mid R, \boldsymbol{c}) = 1$. Following our previous example of the CEO-of relation, $\mathbf{R}_2$ evaluates the likelihood of an organisation to be a company with a CEO position. Importantly, $\mathbf{R}_1$ and $\mathbf{R}_2$ are representations of the relation $R$ and independent of the entities. Therefore, we consider $(\mathbf{R}_1$ and $\mathbf{R}_2)$ to collectively represent the embedding of $R$. Orthogonality of $\mathbf{R}_1, \mathbf{R}_2$ is a requirement for the mathematical proof and also act as a regularisation constraint to prevent overfitting by restricting the relational embedding space. We first perform our mathematical analysis for relational embeddings represented by orthogonal matrices and discuss later how this requirement can be relaxed.

We assume a *slow* random walk where the knowledge vectors do not change significantly between consecutive time steps $(\boldsymbol{c}_k \approx \boldsymbol{c}_{k+1})$. More specifically, we assume that $\|\boldsymbol{c}_k - \boldsymbol{c}_{k+1}\| \leq \epsilon_2$ for some small $\epsilon_2 > 0$. This is a realistic assumption for generating the two entity arguments in the same relational triple because, if the knowledge vectors were significantly different in the two generation steps, then it is likely that the corresponding relations are also different, which would not be coherent with the above-described generative process. Moreover, we assume that the knowledge vectors are distributed uniformly in the unit sphere and denote the distribution of knowledge vectors by $\mathcal{C}$.

To learn KGEs, we must estimate the probability that $h$ and $t$ satisfy the relation $R$, $p(h, t \mid R)$, which can be obtained by taking the expectation of $p(h, t \mid R, \boldsymbol{c}, \boldsymbol{c}')$ w.r.t. $\boldsymbol{c}, \boldsymbol{c}' \sim \mathcal{C}$ given by (3).

$$p(h, t \mid R) = \mathbb{E}_{\boldsymbol{c}, \boldsymbol{c}'}\left[p(h, t \mid R, \boldsymbol{c}, \boldsymbol{c}')\right] \tag{3}$$

$$= \mathbb{E}_{\boldsymbol{c}, \boldsymbol{c}'}\left[p(h \mid R, \boldsymbol{c})p(t \mid R, \boldsymbol{c}')\right] \tag{4}$$

$$= \mathbb{E}_{\boldsymbol{c}, \boldsymbol{c}'}\left[\frac{\exp\left(\boldsymbol{h}^\top \mathbf{R}_1 \boldsymbol{c}\right)}{Z_c} \frac{\exp\left(\boldsymbol{t}^\top \mathbf{R}_2 \boldsymbol{c}'\right)}{Z_{c'}}\right]. \tag{5}$$

Here, partition functions are given by $Z_c = \sum_{h \in \mathcal{V}} \sum_{c \in \mathcal{C}} \exp\left(\boldsymbol{h}^\top \mathbf{R}_1 \boldsymbol{c}\right)$ and $Z_{c'} = \sum_{t \in \mathcal{V}} \sum_{c' \in \mathcal{C}} \exp\left(\boldsymbol{t}^\top \mathbf{R}_2 \boldsymbol{c}'\right)$. (4) follows from our two-step generative process where the generation of $h$ and $t$ in each step is independent given the relation and the corresponding knowledge vectors.

Computing the expectation in (5) is generally difficult because of the two partition functions $Z_c$ and $Z_{c'}$. However, Lemma 1 shows that the partition functions are narrowly distributed around a constant value for all $c$ (or $c'$) values with high probability.

---

[1]We can consider different vocabularies for the entities that can fill the first argument and second argument of relations in a knowledge graph. However, for simplicity, we use a common vocabulary here.

**Lemma 1** (**Concentration Lemma**). *If the entity embedding vectors satisfy the Bayesian prior $v = s\hat{v}$, where $\hat{v}$ is from the spherical Gaussian distribution, and $s$ is a scalar random variable, which is always bounded by a constant $\kappa$, then the entire ensemble of entity embeddings satisfies that*

$$\Pr_{c\sim\mathcal{C}}[(1-\epsilon_z)Z \leq Z_c \leq (1+\epsilon_z)Z] \geq 1-\delta, \tag{6}$$

*for $\epsilon_z = O(1/\sqrt{n})$, and $\delta = \exp(-\Omega(\log^2 n))$, where $n \geq d$ is the number of words and $Z_c$ is the partition function for $c$ given by $\sum_{c\in\mathcal{V}} \exp\left(\boldsymbol{h}^\top \mathbf{R}_1 \boldsymbol{c}\right)$.*

*proof:* To prove the concentration lemma, we show that the mean $\mathbb{E}_{\boldsymbol{h}}[Z_c]$ of $Z_c$ is concentrated around a constant for all knowledge vectors $\boldsymbol{c}$ and its variance is bounded. Recall that

$$Z_c = \sum_{h\in\mathcal{V}} \exp\left(\boldsymbol{h}^\top \mathbf{R}_1 \boldsymbol{c}\right). \tag{7}$$

If $\mathbf{P}$ is an orthogonal matrix and $\boldsymbol{x}$ is a vector, then $\|\mathbf{P}^\top \boldsymbol{x}\|_2^2 = (\mathbf{P}^\top \boldsymbol{x})^\top (\mathbf{P}^\top \boldsymbol{x}) = \boldsymbol{x}^\top \mathbf{P}\mathbf{P}^\top \boldsymbol{x} = \|\boldsymbol{x}\|_2^2$, because $\mathbf{P}^\top \mathbf{P} = \mathbf{I}$. Therefore, from (7) and the orthogonality of the relational embeddings, we see that $\mathbf{R}_1 \boldsymbol{c}$ is a simple rotation of $\boldsymbol{c}$ and does not alter the length of $\boldsymbol{c}$. We represent $\boldsymbol{h} = s_h \hat{\boldsymbol{h}}$, where $s_h = \|\boldsymbol{h}\|$ and $\hat{\boldsymbol{h}}$ is a unit vector (i.e. $\|\hat{\boldsymbol{h}}\|_2 = 1$) distributed on the spherical Gaussian with zero mean and unit covariance matrix $\mathbf{I}_d \in \mathbb{R}^{d\times d}$. Let $s$ be a random variable that has the same distribution as $s_h$. Moreover, let us assume that $s$ is upper bounded by a constant $\kappa$ such that $s \leq \kappa$. From the assumption of the knowledge vector $\boldsymbol{c}$, it is on the unit sphere as well, which is then rotated by $\mathbf{R}_1$.

We can write the partition function using the inner-product between two vectors $\boldsymbol{h}$ and $\mathbf{R}_1 \boldsymbol{c}$, $Z_c = \sum_{h\in\mathcal{V}} \exp\left(\boldsymbol{h}^\top (\mathbf{R}_1 \boldsymbol{c})\right)$. Arora et al. (2016a) showed that (Lemma 2.1 in their paper) the expectation of a partition function of this form can be approximated as follows:

$$\mathbb{E}_{\boldsymbol{c}}[Z_c] = n\mathbb{E}_{\boldsymbol{c}}[\exp\left(\boldsymbol{h}^\top \mathbf{R}_1 \boldsymbol{c}\right)] \tag{8}$$

$$\geq n\mathbb{E}_{\boldsymbol{c}}[1 + \boldsymbol{h}^\top \mathbf{R}_1 \boldsymbol{c}] = n. \tag{9}$$

where $n = |\mathcal{V}|$ is the number of entities in the vocabulary. (8) follows from the expectation of a sum and the independence of $\boldsymbol{h}$ and $\mathbf{R}_1$ from $\boldsymbol{c}$. The inequality of (9) is obtained by applying the Taylor expansion of the exponential series and the final equality is due to the symmetry of the spherical Gaussian. From the law of total expectation, we can write

$$\mathbb{E}_{\boldsymbol{c}}[Z_c] = n\mathbb{E}_{\boldsymbol{c}}[\exp\left(\boldsymbol{h}^\top \mathbf{R}_1 \boldsymbol{c}\right)] = n\mathbb{E}_{s_h}\left[\mathbb{E}_{x|s_h}\left[\exp\left(\boldsymbol{h}^\top \mathbf{R}_1 \boldsymbol{c}\right) \mid s_h\right]\right]. \tag{10}$$

where, $x = \boldsymbol{h}^\top \mathbf{R}_1 \boldsymbol{c}$. Note that conditioned on $s_h$, $\boldsymbol{h}$ is a Gaussian random variable with variance $\sigma^2 = s_h^2$. Therefore, conditioned on $s_h$, $x$ is a random variable with variance $\sigma^2 = \sigma_h^2$. Using this distribution, we can evaluate $\mathbb{E}_{x|s_h}\left[\exp\left(\boldsymbol{h}^\top \mathbf{R}_1 \boldsymbol{c}\right)\right]$ as follows:

$$\mathbb{E}_{x|s_h}\left[\exp\left(\boldsymbol{h}^\top \mathbf{R}_1 \boldsymbol{c}\right) \mid s_h\right] = \int_x \frac{1}{\sqrt{2\pi\sigma^2}} \exp\left(-\frac{x^2}{2\sigma^2}\right) \exp(x) dx \tag{11}$$

$$= \int_x \frac{1}{\sqrt{2\pi\sigma^2}} \exp\left(-\frac{(x-\sigma^2)^2}{2\sigma^2} + \sigma^2/2\right) dx \tag{12}$$

$$= \exp(\sigma^2/2). \tag{13}$$

Therefore, it follows that

$$\mathbb{E}_{\boldsymbol{c}}[Z_c] = n\mathbb{E}_{s_h}[\exp(\sigma^2/2)] = n\mathbb{E}_{s_h}[\exp(s_h^2/2)] = n\exp(s^2/2), \tag{14}$$

where $s$ is the variance of the $\ell_2$ norms of the entity embeddings. Because the set of entities is given and fixed, both $n$ and $\sigma$ are constants, proving that $\mathbb{E}[Z_c]$ does not depend on $c$.

Next, we calculate the variance $\mathbb{V}_{\boldsymbol{c}}[Z_c]$ as follows:

$$\mathbb{V}_{\boldsymbol{c}}[Z_c] = \sum_h \mathbb{V}_{\boldsymbol{c}}[\exp\left(\boldsymbol{h}^\top \mathbf{R}_1 \boldsymbol{c}\right)]$$

$$\leq n\mathbb{E}_{\boldsymbol{c}}\left[\exp\left(\boldsymbol{h}^\top \mathbf{R}_1 \boldsymbol{c}\right)\right]$$

$$= n\mathbb{E}_{s_h}\left[\mathbb{E}_{x|s_h}\left[\exp\left(2\boldsymbol{h}^\top \mathbf{R}_1 \boldsymbol{t}\right) \mid s_h\right]\right]. \tag{15}$$

Because $2\boldsymbol{h}^\top \mathbf{R}_1 \boldsymbol{t}$ is a Gaussian random variable with variance $4\sigma^2 = 4s_h^2$ from a similar calculation as in (11) we obtain,

$$\mathbb{E}_{x|s_h}\left[\exp\left(2\boldsymbol{h}^\top \mathbf{R}_1 \boldsymbol{t}\right) \mid s_h\right] = \exp(2\sigma^2). \tag{16}$$

By substituting (16) in (15) we have that

$$\mathbb{V}_{\boldsymbol{c}}[Z_c] \le n\mathbb{E}_{s_h}\left[\exp\left(2\sigma^2\right)\right] = n\mathbb{E}_{s_h}\left[\exp(2s^2)\right] \le \Lambda n \tag{17}$$

for $\Lambda = \exp(8\kappa^2)$ a constant bounding $s \le \kappa$ as stated. $\qquad\square$

From above, we have bounded both the mean and variance of the partition function by constants that are independent of the knowledge vector. Note that neither $\exp\left(\boldsymbol{h}^\top \mathbf{R}_1 \boldsymbol{c}\right)$ nor $\exp\left(\boldsymbol{t}^\top \mathbf{R}_2 \boldsymbol{c}'\right)$ are sub-Gaussian nor sub-exponential. Therefore, standard concentration bounds derived for sub-Gaussian or sub-exponential random variables cannot be used in our analysis. However, the argument given in Appendix A.1 in Arora et al. (2016b) for a partition function with bounded mean and variance can be directly applied to $Z_c$ in our case, which completes the proof of the concentration lemma. $\qquad\square$

From the symmetry between $h$ and $t$, Lemma 1 also applies for the partition function $\sum_{t \in \mathcal{V}} \left(\boldsymbol{t}^\top \mathbf{R}_2 \boldsymbol{c}'\right)$. Under the conditions required to satisfy Lemma 1, the following main theorem of this paper holds:

**Theorem 1.** *Suppose that the entity embeddings satisfy* (1)*. Then, we have*

$$\log p(h, t \mid R) = \frac{\|\mathbf{R}_1{}^\top \boldsymbol{h} + \mathbf{R}_2{}^\top \boldsymbol{t}\|_2^2}{2d} - 2\log Z \pm \epsilon. \tag{18}$$

*for* $\epsilon = O(1/\sqrt{n}) + \widetilde{O}(1/d)$*, where*

$$Z = Z_c = Z_{c'}. \tag{19}$$

The complete proof of Theorem 1 is given in Appendix A. Below we briefly sketch the main steps.

*Proof sketch:* Let $F$ be the event that both $c$ and $c'$ are within $(1 \pm \epsilon_z)Z$. Then, from Lemma 1 and the union bound, event $F$ happens with probability at least $1 - 2\exp(-\Omega(\log^2 n))$. The R.H.S. of (5) can be split into two parts $T_1$ and $T_2$ according to whether $F$ happens or not.

$$p(h, t \mid R) = \underbrace{\mathbb{E}_{c,c'}\left[\frac{\exp\left(\boldsymbol{h}^\top \mathbf{R}_1 \boldsymbol{c}\right)}{Z_c} \frac{\exp\left(\boldsymbol{h}^\top \mathbf{R}_2 \boldsymbol{c}'\right)}{Z_{c'}} \mathbf{1}_F\right]}_{=T_1} + \underbrace{\mathbb{E}_{c,c'}\left[\frac{\exp\left(\boldsymbol{h}^\top \mathbf{R}_1 \boldsymbol{c}\right)}{Z_c} \frac{\exp\left(\boldsymbol{h}^\top \mathbf{R}_2 \boldsymbol{c}'\right)}{Z_{c'}} \mathbf{1}_{\bar{F}}\right]}_{=T_2}. \tag{20}$$

$T_1$ can be approximated as given by (21).

$$T_1 = \frac{1 \pm \mathcal{O}(\epsilon_z)}{Z^2} \mathbb{E}_{c,c'}\left[\exp\left(\boldsymbol{h}^\top \mathbf{R}_1 \boldsymbol{c}\right) \exp\left(\boldsymbol{t}^\top \mathbf{R}_2 \boldsymbol{c}'\right)\right] \tag{21}$$

On the other hand, $T_2$ can be shown to be a constant, independent of $d$, given by (22).

$$|T_2| = \exp(-\Omega(\log^{1.8} n)) \tag{22}$$

The vocabulary size $n$ of real-world knowledge graphs is typically over $10^5$, for which $T_2$ becomes negligibly small. Therefore, it suffices to consider only $T_1$. Because of the slowness of the random walk we have $\boldsymbol{c} \approx \boldsymbol{c}'$

Using the law of total expectation we can write $T_1$ as follows:

$$T_1 = \frac{1 \pm \mathcal{O}(\epsilon_z)}{Z^2} \mathbb{E}_c\left[\exp\left(\boldsymbol{h}^\top \mathbf{R}_1 \boldsymbol{c}\right) \mathbb{E}_{c'|c}\left[\exp\left(\boldsymbol{t}^\top \mathbf{R}_2 \boldsymbol{c}'\right)\right]\right]$$

$$= \frac{1 \pm \mathcal{O}(\epsilon_z)}{Z^2} \mathbb{E}_c\left[\exp\left(\boldsymbol{h}^\top \mathbf{R}_1 \boldsymbol{c}\right) A(c)\right] \tag{23}$$

where $A(c) \coloneqq \mathbb{E}_{c'|c}\left[\exp\left(\boldsymbol{t}^\top \mathbf{R}_2 \boldsymbol{c}'\right)\right]$. Doing some further evaluations we show that

$$A(c) = (1 \pm \epsilon_2)\exp\left(\boldsymbol{t}^\top \mathbf{R}_2 \boldsymbol{c}\right) \tag{24}$$

Plugging (50) back in (23) provides the claim of the theorem. $\qquad\square$

The relationship given by (18) indicates that head and tail entity embeddings are first transformed respectively by $\mathbf{R}_1{}^\top$ and $\mathbf{R}_2{}^\top$, and the squared $\ell_2$ norm of the sum of the transformed vectors is proportional to the probability $p(h, t \mid R)$.

## 3 LEARNING KNOWLEDGE GRAPH EMBEDDINGS

In this section, we derive a training objective from Theorem 1 that we can then optimise to learn KGE. The goal is to empirically validate the theoretical result by evaluating the learnt KGEs. Knowledge graphs represent information about relations between two entities in the form of *relational triples*. The joint probability $p(h, R, t)$ given by Theorem 1 is useful for determining whether a relation $R$ exists between two given entities $h$ and $t$. For example, if we know that with a high probability that $R$ holds between $h$ and $t$, then we can append $(h, R, t)$ to the knowledge graph. The task of expanding knowledge graphs by predicting missing links between entities or relations is known as the *link prediction* problem (Trouillon et al., 2016). In particular, if we can automatically append such previously unknown knowledge to the knowledge graph, we can expand the knowledge graph and address the knowledge acquisition bottleneck.

To derive a criteria for determining whether a link must be predicted among entities and relations, let us consider a relational triple $(h, R, t) \in \mathcal{D}$ that exists in a given knowledge graph $\mathcal{D}$. We call such relational triples as *positive* triples because from the assumption it is known that $R$ holds between $h$ and $t$. On the other hand, consider a *negative* relational triple $(h', R, t') \in \mathcal{D}$ formed by, for example, randomly perturbing a positive triple. A popular technique for generating such (pseudo) negative triples is to replace $h$ or $t$ with a randomly selected different instance of the same entity type. As an alternative for random perturbation, Cai and Wang (2018) proposed a method for generating negative instances using adversarial learning. Here, we are not concerned about the actual method used for generating the negative triples but assume a set of negative triples, $\bar{\mathcal{D}}$, generated using some method, to be given.

Given a positive triple $(h, R, t) \in \mathcal{D}$ and a negative triple $(h', R, t') \in \bar{\mathcal{D}}$, we would like to learn KGEs such that a higher probability is assigned to $(h, R, t)$ than that assigned to $(h', R, t')$. We can formalise this requirement using the likelihood ratio given by (25).

$$\frac{p(h, R, t)}{p(h', R, t')} \geq \eta \tag{25}$$

Here, $\eta > 1$ is a threshold that determines how higher we would like to set the probabilities for the positive triples compares to that of the negative triples.

By taking the logarithm of both sides in (25) we obtain

$$\log p(h, R, t) - \log p(h', R, t') \geq \log \eta$$
$$\log \eta + \log p(h', R, t') - \log p(h, R, t) \geq 0 \tag{26}$$

If a positive triple $(h, R, t)$ is correctly assigned a higher probability than a negative triple $p(h', R, t')$, then the left hand side of (26) will be negative, indicating that there is no *loss* incurred during this classification task. Therefore, we can re-write (26) to obtain the *marginal loss* Bordes et al. (2013; 2011), $L(\mathcal{D}, \bar{\mathcal{D}})$, a popular choice as a learning objective in prior work in KGE, as shown in (27).

$$L(\mathcal{D}, \bar{\mathcal{D}}) = \sum_{\substack{(h, R, t) \in \mathcal{D} \\ (h', R, t') \in \bar{\mathcal{D}}}} \max\left(0, \log \eta + \log p(h', R, t') - \log p(h, R, t)\right)$$

$$= \max\left(0, 2d \log \eta + \|\mathbf{R}_1^\top \boldsymbol{h}' + \mathbf{R}_2^\top \boldsymbol{t}'\|_2^2 - \|\mathbf{R}_1^\top \boldsymbol{h} + \mathbf{R}_2^\top \boldsymbol{t}\|_2^2\right) \tag{27}$$

We can assume $2d \log \eta$ to be the *margin* for the constraint violation.

Theorem 1 requires $\mathbf{R}_1$ and $\mathbf{R}_2$ to be orthogonal. To reflect this requirement, we add two $\ell_2$ regularisation terms $\|\mathbf{R}_1^\top \mathbf{R}_1 - \mathbf{I}\|_2^2$ and $\|\mathbf{R}_2^\top \mathbf{R}_2 - \mathbf{I}\|_2^2$ respectively with regularisation coefficients $\lambda_1$ and $\lambda_2$ to the objective function given by (27). In our experiments, we compute the gradients (27) w.r.t. each of the parameters $\boldsymbol{h}, \boldsymbol{t}, \boldsymbol{R}_1$ and $\boldsymbol{R}_2$ and use stochastic gradient descent (SGD) for optimisation. This approach can be easily extended to learn from multiple negative triples as shown in Appendix B.

## 4 RELATED WORK

At a high-level of abstraction, KGE methods can be seen as differing in their design choices for the following two main problems: (a) how to represent entities and relations, and (b) how to model the

interaction between two entities and a relation that holds between them. Next, we briefly discuss prior proposals to those two problems (refer (Wang et al., 2017; Nickel et al., 2015; Nguyen, 2017) for an extended survey on KGE).

A popular choice for representing entities is to use vectors, whereas relations have been represented by vectors, matrices or tensors. For example, TransE (Bordes et al., 2011), TransH (Wang et al., 2014), TransD (Ji et al., 2015), TransG (Xiao et al., 2016), TransR (Lin et al., 2015), lppTransD (Yoon et al., 2016), DistMult (Yang et al., 2015), HolE (Nickel et al., 2016) and ComplEx (Trouillon et al., 2016) represent relations by vectors, whereas Structured Embeddings (Bordes et al., 2011), TranSparse (Ji et al., 2016), STransE (Nguyen et al., 2016), RESCAL (Nickel et al., 2011) use matrices and Neural Tensor Network (NTN) (Socher et al., 2013) uses 3D tensors. ComplEx (Trouillon et al., 2016) introduced complex vectors for KGEs to capture the asymmetry in semantic relations. (Ding et al., 2018) obtained state-of-the-art performance for KGE by imposing non-negativity and entailment constraints to ComplEx.

Given entity and relation embeddings, a scoring function is defined that evaluates the strength of a relation $R$ between two entities $h$ and $t$ in a triple $(h, R, t)$. The scoring functions that encode various intuitions have been proposed such as the $\ell_1$ or $\ell_2$ norms of the vector formed by a translation of the head entity embedding by the relation embedding over the target embedding, or by first performing a projection from the entity embedding space to the relation embedding space (Yoon et al., 2016) As an alternative to using vector norms as scoring functions, DistMult and ComplEx use the component-wise multi-linear dot product.

Once a scoring function is defined, KGEs are learnt that assign better scores to relational triples in existing knowledge graphs (positive triples) over triples where the relation does not hold (negative triples) by minimising a loss function such as the logistic loss (RESCAL, DistMult, ComplEx) or marginal loss (TransE, TransH, TransD, TransD). Because knowledge graphs record only positive triples, a popular method to generate pseudo negative triples is to perturb a positive instance by replacing its head or tail entity by an entity selected uniformly at random from the vocabulary of the entities. However, uniformly sampled negative triples are likely to be obvious examples that do not provide much information to the learning process and can be detected by simply checking for the type of the entities in a triple. Cai and Wang (2018) proposed an adversarial learning approach where a *generator* assigns a probability to each relation triple and negative instances are sampled according to this probability distribution to train a *discriminator* that discriminates between positive and negative instances. (Xiao et al., 2016) proposed TransG, a generative model based on the Chinese restaurant process, to model multiple relations that exist between a pair of entities. However, their relation embeddings are designed to satisfy vector translation similar to TransE.

As an alternative to directly learning embeddings from a graph, several methods (Grover and Leskovec, 2016; Perozzi et al., 2014; Ristoski et al., 2018) have considered the vertices visited during truncated random walks over the graph as *pseudo sentences*, and have applied popular word embedding learning algorithms such as skip-gram with negative sampling or continuous bag-of-words model (Mikolov et al., 2013) to learn vertex embeddings. However, pseudo sentences generated this way are syntactically very different from sentences in natural languages.

On the other hand, our work extends the random walk analysis by Arora et al. (2016a) that derives a useful connection between the joint co-occurrence probability of two words and the $\ell_2$ norm of the sum of the corresponding word embeddings. Specifically, they proposed a latent variable model where the words in a corpus are generated by a probabilistic model parametrised by a time-dependent discourse vector that performs a random walk. However, unlike in our work, they do not consider the relations between two co-occurring words in a corpus. Bollegala et al. (2018) extended the model proposed by Arora et al. (2016a) to capture co-occurrences involving more than two words. They defined the co-occurrence of $k$ unique words in a given context as a $k$-way co-occurrence, where Arora et al. (2016a)'s result could be seen as a special case coresponding to $k = 2$. Moreover, Bollegala et al. (2018) showed that it is possible to learn word embeddings that capture some types of semantic relations such as antonymy and collocation using 3-way co-occurrences more accurately than using 2-way co-occurrences. However, their model does not explicitly consider the relations between words/entities and uses only a corpus for learning the word embeddings.

Table 2: Triple classification.

| Method | Accuracy | |
|--------|------|------|
|        | WN11 | FB13 |
| SE      | 53.0  | 75.2 |
| TransE  | 75.9  | 81.5 |
| TransR  | 85.9  | 82.5 |
| TransG  | **87.4** | 87.3 |
| NTN     | 70.4  | 87.1 |
| RelWalk | 75.48 | **87.5** |

Table 3: Link prediction. Results marked with [⋆] are taken from Dettmers et al. (2017), [●] from Nguyen et al. (2017), [◁] from and Cai and Wang (2018). All other results for the baselines are taken from their original papers.

| Method | FB15K237 | | | | | WN18RR | | | | |
|--------|-----|-----|-----|-----|------|-----|-----|-----|-----|------|
|        | MRR | MR | H@1 | H@3 | H@10 | MRR | MR | H@1 | H@3 | H@10 |
| TransE● | 0.294 | 347 | - | - | 0.465 | 0.226 | 3384 | - | - | 0.50 |
| TransD◁ | 0.28 | - | - | - | 0.453 | - | - | - | - | 0.43 |
| DistMult⋆ | 0.241 | 254 | 0.155 | 0.263 | 0.419 | 0.43 | 5110 | 0.39 | 0.44 | 0.49 |
| ComplEx⋆ | 0.247 | 339 | 0.158 | 0.275 | 0.428 | 0.44 | 5261 | 0.41 | 0.46 | **0.51** |
| ConvE | 0.316 | 246 | 0.239 | 0.35 | 0.491 | **0.46** | 5277 | 0.39 | 0.43 | 0.48 |
| RelWalk | **0.329** | **105** | **0.243** | **0.354** | **0.502** | 0.451 | **3232** | **0.42** | **0.47** | **0.51** |

## 5 EMPIRICAL VALIDATION

To empirically evaluate the theoretical result stated in Theorem 1, we learn KGEs (denoted by **RelWalk**) by minimising the marginal loss objective derived in section 3. We use the FB15k237, FB13 (subsets of *Freebase*) and WN11, WN18RR (subsets of *WordNet*) datasets, which are standard benchmarks for KGE. We use the standard training, validation and test splits as detailed in Table 4. We generate negative triples by replacing a head or a tail entity in a positive triple by a randomly selected different entity and learn KGEs. We train the model until convergence or at most 1000 epochs over the training data where each epoch is divided into 100 mini-batches. The best model is selected by early stopping based on the performance of the learnt embeddings on the validation set (evaluated after each 20 epochs). The training details and hyperparameter settings are detailed in Appendix C. **RelWalk** is implemented in the open-source toolkit OpenKE (Han et al., 2018).[2]

We conduct two evaluation tasks: *link prediction* (predict the missing head or tail entity in a given triple $(h, R, ?)$ or $(?, R, t)$) (Bordes et al., 2011) and *triple classification* (predict whether a relation $R$ holds between $h$ and $t$ in a given triple $(h, R, t)$) (Socher et al., 2013). We evaluate the performance in the link prediction task using mean reciprocal rank (**MRR**), mean rank (**MR** (the average of the rank assigned to the original head or tail entity in a corrupted triple) and hits at ranks 1, 3 and 10 (**H@1,3,10**), whereas in the triple classification task we use **accuracy** (percentage of the correctly classified test triples). We only report scores under the *filtered* setting Bordes et al. (2013), which removes all triples appeared in training, validating and testing sets from candidate triples before obtaining the rank of the ground truth triple. In link prediction, we consider all entities that appear in the corresponding argument in the entire knowledge graph as candidates.

In Tables 2 and 3 we compare the KGEs learnt by **RelWalk** against prior work using the published results. For link prediction, **RelWalk** reports SoTA on both WN18RR and FB15K237 in all evaluation measures, except against ConvE in WN18RR measured by MRR. WN18RR excludes triples from WN18 that are simply inverted between train and test partitions (Toutanova and Chen, 2015; Dettmers et al., 2017). **RelWalk**'s consistently good performance on both versions of this dataset shows that it is considering the global structure in the knowledge graph when learning KGEs. For triple classification, **RelWalk** reports the best performance on FB13, whereas TransG reports the best performance on

---

[2]To facilitate the double blind policy, the source code for RelWalk will be released upon paper acceptance

WN11. Considering that both TransG and **RelWalk** are generative models, it would be interesting to further investigate generative approaches for KGE in the future. Overall, the experimental results support our theoretical claim and emphasise the importance of theoretically motivating the scoring function design process.

## 6 CONCLUSION

We proposed **RelWalk**, a generative model of KGE and derived a theoretical relationship between the probability of a triple and entity, relation embeddings. We then proposed a learning objective based on the theoretical relationship we derived. Experimental results on a link prediction and a triple classification tasks show that **RelWalk** obtains strong performances in multiple benchmark datasets.

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

APPENDIX

## A    PROOF OF THEOREM 1

Let us consider the probabilistic event that $(1 - \epsilon_z)Z \leq Z_c \leq (1 + \epsilon_z)Z$ to be $F_c$ and $(1 - \epsilon_z)Z \leq Z_{c'} \leq (1 + \epsilon_z)Z$ to be $F_{c'}$. From Lemma 1 we have $\Pr_c[F_c] \geq 1 - \delta$. Then from the union bound we have,

$$
\begin{aligned}
\Pr[\bar{F}_c \vee \bar{F}_{c'}] &\leq \Pr[\bar{F}_c] + \Pr[\bar{F}_{c'}] \\
&= 1 - \Pr[F_c] + 1 - \Pr[F_{c'}] \\
&= 2\delta.
\end{aligned}
\tag{28}
$$

Moreover, let $F$ be the probabilistic event that both $F_c$ and $F_{c'}$ being True. Then from $\Pr[F] = 1 - \Pr[\bar{F}_c \vee \bar{F}_{c'}]$ we have, $\Pr[F] \geq 1 - 2\delta$. We can decompose the expectation in the R.H.S. in (5) into two terms $T_1$ and $T_2$ depending on whether respectively $F$ is True or False as follows:

$$
p(h, t \mid r) = \underbrace{\mathbb{E}_{c,c'}\left[\frac{\exp\left(\boldsymbol{h}^\top \mathbf{R}_1 \boldsymbol{c}\right)}{Z_c} \frac{\exp\left(\boldsymbol{h}^\top \mathbf{R}_2 \boldsymbol{c}'\right)}{Z_{c'}} \mathbf{1}_F\right]}_{=T_1} + \underbrace{\mathbb{E}_{c,c'}\left[\frac{\exp\left(\boldsymbol{h}^\top \mathbf{R}_1 \boldsymbol{c}\right)}{Z_c} \frac{\exp\left(\boldsymbol{h}^\top \mathbf{R}_2 \boldsymbol{c}'\right)}{Z_{c'}} \mathbf{1}_{\bar{F}}\right]}_{=T_2}.
\tag{29}
$$

Here, $\mathbf{1}_F$ and $\mathbf{1}_{\bar{F}}$ are indicator functions given by:

$$
\mathbf{1}_F = \begin{cases} 1 & \text{if } F \text{ is True}, \\ 0 & \text{otherwise}, \end{cases}
\tag{30}
$$

and

$$
\mathbf{1}_{\bar{F}} = \begin{cases} 0 & \text{if } F \text{ is True}, \\ 1 & \text{otherwise}. \end{cases}
\tag{31}
$$

Let us first show that $T_2$ is negligibly small.

For two real integrable functions $\psi_1(x)$ and $\psi_2(x)$ in $[a, b]$, the Cauchy-Schwarz's inequality states that

$$
\left[\int_a^b \psi_1(x)\psi_2(x)dx\right]^2 \leq \int_a^b [\psi_1(x)]^2 dx \int_a^b [\psi_2(x)]^2 dx.
\tag{32}
$$

Applying (32) to $T_2$ in (29) we have:

$$
\begin{aligned}
&\left(\mathbb{E}_{c,c'}\left[\frac{1}{Z_c Z_{c'}} \exp\left(\boldsymbol{h}^\top \mathbf{R}_1 \boldsymbol{c}\right) \exp\left(\boldsymbol{t}^\top \mathbf{R}_2 \boldsymbol{c}'\right) \mathbf{1}_{\bar{F}}\right]\right)^2 \\
&\leq \left(\mathbb{E}_{c,c'}\left[\frac{1}{Z_c^2} \exp\left(\boldsymbol{h}^\top \mathbf{R}_1 \boldsymbol{c}\right)^2 \mathbf{1}_{\bar{F}}\right]\right)\left(\mathbb{E}_{c,c'}\left[\frac{1}{Z_{c'}^2} \exp\left(\boldsymbol{t}^\top \mathbf{R}_2 \boldsymbol{c}'\right)^2 \mathbf{1}_{\bar{F}}\right]\right) \\
&= \left(\mathbb{E}_c\left[\frac{1}{Z_c^2} \exp\left(\boldsymbol{h}^\top \mathbf{R}_1 \boldsymbol{c}\right)^2 \mathbb{E}_{c'|c}\left[\mathbf{1}_{\bar{F}}\right]\right]\right)\left(\mathbb{E}_{c'}\left[\frac{1}{Z_{c'}^2} \exp\left(\boldsymbol{t}^\top \mathbf{R}_2 \boldsymbol{c}'\right)^2 \mathbb{E}_{c|c'}\left[\mathbf{1}_{\bar{F}}\right]\right]\right)
\end{aligned}
\tag{33}
$$

Note that $Z_c \geq 1$ because $Z_c$ is the sum of positive numbers and if $\boldsymbol{h}^\top \mathbf{R}_1 \boldsymbol{c} \geq 0$ for at least one of the $h \in \mathcal{V}$, then the total sum will be greater than 1. Therefore, by dropping $Z_c$ term from the denominator we can further increase the first term in (33) as given by (34).

$$
\mathbb{E}_c\left[\frac{1}{Z_c^2} \exp\left(\boldsymbol{h}^\top \mathbf{R}_1 \boldsymbol{c}\right)^2 \mathbb{E}_{c'|c}\left[\mathbf{1}_{\bar{F}}\right]\right] \leq \mathbb{E}_c\left[\exp\left(\boldsymbol{h}^\top \mathbf{R}_1 \boldsymbol{c}\right)^2 \mathbb{E}_{c'|c}\left[\mathbf{1}_{\bar{F}}\right]\right]
\tag{34}
$$

Let us split the expectation on the R.H.S. of (34) into two cases depending on whether $\boldsymbol{h}^\top \mathbf{R}_1 \boldsymbol{c} > 0$ or otherwise, indicated respectively by $\mathbf{1}_{(\boldsymbol{h}^\top \mathbf{R}_1 \boldsymbol{c} > 0)}$ and $\mathbf{1}_{(\boldsymbol{h}^\top \mathbf{R}_1 \boldsymbol{c} \leq 0)}$.

$$
\begin{aligned}
&\mathbb{E}_c\left[\exp\left(\boldsymbol{h}^\top \mathbf{R}_1 \boldsymbol{c}\right)^2 \mathbb{E}_{c'|c}\left[\mathbf{1}_{\bar{F}}\right]\right] \\
&= \mathbb{E}_c\left[\exp\left(\boldsymbol{h}^\top \mathbf{R}_1 \boldsymbol{c}\right)^2 \mathbf{1}_{(\boldsymbol{h}^\top \mathbf{R}_1 \boldsymbol{c} > 0)} \mathbb{E}_{c'|c}\left[\mathbf{1}_{\bar{F}}\right]\right] + \mathbb{E}_c\left[\exp\left(\boldsymbol{h}^\top \mathbf{R}_1 \boldsymbol{c}\right)^2 \mathbf{1}_{(\boldsymbol{h}^\top \mathbf{R}_1 \boldsymbol{c} \leq 0)} \mathbb{E}_{c'|c}\left[\mathbf{1}_{\bar{F}}\right]\right]
\end{aligned}
\tag{35}
$$

The second term of (35) is upper bounded by

$$\mathbb{E}_{c,c'}\left[\mathbf{1}_{\bar{F}}\right] \leq \exp\left(-\Omega(\log^2 n)\right) \tag{36}$$

The first term of (35) can be bounded as follows:

$$\mathbb{E}_c\left[\exp\left(\boldsymbol{h}^\top \mathbf{R}_1 \boldsymbol{c}\right)^2 \mathbf{1}_{(\boldsymbol{h}^\top \mathbf{R}_1 \boldsymbol{c}>0)} \mathbb{E}_{c'|c}\left[\mathbf{1}_{\bar{F}}\right]\right] \leq \mathbb{E}_c\left[\exp(\alpha \boldsymbol{h}^\top \mathbf{R}_1 \boldsymbol{c})^2 \mathbf{1}_{(\boldsymbol{h}^\top \mathbf{R}_1 \boldsymbol{c}>0)} \mathbb{E}_{c'|c}\left[\mathbf{1}_{\bar{F}}\right]\right]$$

$$\leq \mathbb{E}_c\left[\exp(\alpha \boldsymbol{h}^\top \mathbf{R}_1 \boldsymbol{c})^2 \mathbb{E}_{c'|c}\left[\mathbf{1}_{\bar{F}}\right]\right] \tag{37}$$

where $\alpha > 1$. Therefore, it is sufficient to bound $\mathbb{E}_c\left[\exp(\alpha \boldsymbol{h}^\top \mathbf{R}_1 \boldsymbol{c})^2 \mathbb{E}_{c'|c}\left[\mathbf{1}_{\bar{F}}\right]\right]$ when $\|\boldsymbol{h}\| = \Omega(\sqrt{d})$.

Let us denote by $z$ the random variable $2\boldsymbol{h}^\top \mathbf{R}_1 \boldsymbol{c}$. Moreover, let $r(z) = \mathbb{E}_{c'|z}[\mathbf{1}_{\bar{F}}]$, which is a function of $z$ between $[0, 1]$. We wish to upper bound $\mathbb{E}_c[\exp(z)r(z)]$. The worst-case $r(z)$ can be quantified using a continuous version of Abel's inequality (proved as Lemma A.4 in Arora et al. (2016b)), we can upper bound $\mathbb{E}_c\left[\exp(z)r(z)\right]$ as follows:

$$\mathbb{E}_c\left[\exp(z)r(z)\right] \leq \mathbb{E}\left[\exp(z)\mathbf{1}_{[t,+\infty]}(z)\right] \tag{38}$$

where $t$ satisfies that $\mathbb{E}_c[\mathbf{1}_{[t,+\infty]}(z)] = \Pr[z \geq t] = \mathbb{E}_c[r(z)] \leq \exp(-\Omega(\log^2 n))$. Here, $\mathbf{1}_{[t,+\infty]}(z)$ is a function that takes the value 1 when $z \geq t$ and zero elsewhere. Then, we claim $\Pr_c[z \geq t] \leq \exp(-\Omega(\log^2 n))$ implies that $t \geq \Omega(\log^{.9} n)$.

If $c$ was distributed as $\mathcal{N}(0, \frac{1}{d}\mathbf{I})$, this would be a simple tail bound. However, as $c$ is distributed uniformly on the sphere, this requires special care, and the claim follows by applying the tail bound for the spherical distribution given by Lemma A.1 in (Arora et al., 2016a) instead. Finally, applying Corollary A.3 in (Arora et al., 2016a), we have:

$$\mathbb{E}[\exp(z)r(z)] \leq \mathbb{E}[\exp(z)\mathbf{1}_{[t,+\infty]}(z)] = \exp(-\Omega(\log^{1.8} n)) \tag{39}$$

From a similar argument as above we can obtain the same bound for $c'$ as well. Therefore, $T_2$ in (29) can be upper bounded as follows:

$$\mathbb{E}_{c,c'}\left[\frac{1}{Z_c Z_{c'}} \exp\left(\boldsymbol{h}^\top \mathbf{R}_1 \boldsymbol{c}\right) \exp\left(\boldsymbol{t}^\top \mathbf{R}_2 \boldsymbol{c'}\right) \mathbf{1}_{\bar{F}}\right]$$

$$= \left(\mathbb{E}_c\left[\frac{1}{Z_c^2} \exp\left(\boldsymbol{h}^\top \mathbf{R}_1 \boldsymbol{c}\right)^2 \mathbb{E}_{c'|c}\left[\mathbf{1}_{\bar{F}}\right]\right]\right)^{1/2} \left(\mathbb{E}_{c'}\left[\frac{1}{Z_{c'}^2} \exp\left(\boldsymbol{t}^\top \mathbf{R}_2 \boldsymbol{c'}\right)^2 \mathbb{E}_{c|c'}\left[\mathbf{1}_{\bar{F}}\right]\right]\right)^{1/2}$$

$$\leq \exp(-\Omega(\log^{1.8} n)) \tag{40}$$

Because $n = |\mathcal{V}|$, the size of the entity vocabulary, is large (ca. $n > 10^5$) in most knowledge graphs, we can ignore the $T_2$ term in (29). Combining this with (29) we obtain an upper bound for $p(h, t \mid R)$ given by (41).

$$p(h, t \mid R) \leq (1 + \epsilon_z)^2 \frac{1}{Z^2} \mathbb{E}_{c,c'}\left[\exp\left(\boldsymbol{h}^\top \mathbf{R}_1 \boldsymbol{c}\right) \exp\left(\boldsymbol{t}^\top \mathbf{R}_2 \boldsymbol{c'}\right) \mathbf{1}_F\right] + |\mathcal{D}| \exp(-\Omega(\log^{1.8} n))$$

$$= (1 + \epsilon_z)^2 \frac{1}{Z^2} \mathbb{E}_{c,c'}\left[\exp\left(\boldsymbol{h}^\top \mathbf{R}_1 \boldsymbol{c}\right) \exp\left(\boldsymbol{t}^\top \mathbf{R}_2 \boldsymbol{c'}\right)\right] + \delta_0 \tag{41}$$

where $|\mathcal{D}|$ is the number of relational tuples $(h, R, t)$ in the KB $\mathcal{D}$ and $\delta_0 = |\mathcal{D}| \exp(-\Omega(\log^{1.8} n)) \leq \exp(-\Omega(\log^{1.8} n))$ by the fact that $Z \leq \exp(2\kappa)n = O(n)$, where $\kappa$ is the upper bound on $\boldsymbol{h}^\top \mathbf{R}_1 \boldsymbol{c}$ and $\boldsymbol{t}^\top \mathbf{R}_2 \boldsymbol{c'}$, which is regarded as a constant.

On the other hand, we can lower bound $p(h, t \mid R)$ as given by (42).

$$p(h, t \mid R) \geq (1 - \epsilon_z)^2 \frac{1}{Z^2} \mathbb{E}_{c,c'}\left[\exp\left(\boldsymbol{h}^\top \mathbf{R}_1 \boldsymbol{c}\right) \exp\left(\boldsymbol{t}^\top \mathbf{R}_2 \boldsymbol{c'}\right) \mathbf{1}_F\right]$$

$$\geq (1 - \epsilon_z)^2 \frac{1}{Z^2} \mathbb{E}_{c,c'}\left[\exp\left(\boldsymbol{h}^\top \mathbf{R}_1 \boldsymbol{c}\right) \exp\left(\boldsymbol{t}^\top \mathbf{R}_2 \boldsymbol{c'}\right)\right] - |\mathcal{D}| \exp(-\Omega(\log^{1.8} n))$$

$$\geq (1 - \epsilon_z)^2 \frac{1}{Z^2} \mathbb{E}_{c,c'}\left[\exp\left(\boldsymbol{h}^\top \mathbf{R}_1 \boldsymbol{c}\right) \exp\left(\boldsymbol{t}^\top \mathbf{R}_2 \boldsymbol{c'}\right)\right] - \delta_0 \tag{42}$$

Taking the logarithm of both sides, from (41) and (42), the multiplicative error translates to an additive error given by (43).

$$
\begin{aligned}
\log p(h, t \mid R) &= \log \left( \mathbb{E}_{c,c'} \left[ \exp \left( \boldsymbol{h}^\top \mathbf{R}_1 \boldsymbol{c} \right) \exp \left( \boldsymbol{t}^\top \mathbf{R}_2 \boldsymbol{c}' \right) \right] \pm \delta_0 \right) - 2 \log Z + 2 \log(1 \pm \epsilon_z) \\
&= \log \left( \mathbb{E}_c \left[ \exp \left( \boldsymbol{h}^\top \mathbf{R}_1 \boldsymbol{c} \right) \mathbb{E}_{c'|c} \left[ \exp \left( \boldsymbol{t}^\top \mathbf{R}_2 \boldsymbol{c}' \right) \right] \right] \pm \delta_0 \right) - 2 \log Z + 2 \log(1 \pm \epsilon_z) \\
&= \log \left( \mathbb{E}_c \left[ \exp \left( \boldsymbol{h}^\top \mathbf{R}_1 \boldsymbol{c} \right) A(c) \pm \delta_0 \right] \right) - 2 \log Z + 2 \log(1 \pm \epsilon_z) \quad (43)
\end{aligned}
$$

where $A(c) := \mathbb{E}_{c'|c} \left[ \exp \left( \boldsymbol{t}^\top \mathbf{R}_2 \boldsymbol{c}' \right) \right]$.

We assumed that $\boldsymbol{c}$ and $\boldsymbol{c}'$ are on the unit sphere and $\mathbf{R}_1$ and $\mathbf{R}_2$ to be orthogonal matrices. Therefore, $\mathbf{R}_1 \boldsymbol{c}$ and $\mathbf{R}_2 \boldsymbol{c}'$ are also on the unit sphere. Moreover, if we let the upper bound of the $\ell_2$ norm of the entity embeddings to be $\kappa' \sqrt{d}$, then we have $\|\boldsymbol{h}\| \le \kappa' \sqrt{d}$ and $\|\boldsymbol{t}\| \le \kappa' \sqrt{d}$. Therefore, we have

$$
\langle \mathbf{R}_1 \boldsymbol{h}, \boldsymbol{c}' - \boldsymbol{c} \rangle \le \|\boldsymbol{h}\| \|\boldsymbol{c} - \boldsymbol{c}'\| \le \kappa' \sqrt{d} \|\boldsymbol{c} - \boldsymbol{c}'\| \quad (44)
$$

Then we can lower bound $A(c)$ as follows:

$$
\begin{aligned}
A(c) &= \exp \left( \boldsymbol{t}^\top \mathbf{R}_2 \boldsymbol{c} \right) \mathbb{E}_{c'|c} \left[ \exp \left( \boldsymbol{t}^\top \mathbf{R}_2 (\boldsymbol{c}' - \boldsymbol{c}) \right) \right] \\
&\le \exp \left( \boldsymbol{t}^\top \mathbf{R}_2 \boldsymbol{c} \right) \mathbb{E}_{c'|c} \left[ \exp \left( \kappa' \sqrt{d} \|\boldsymbol{c}' - \boldsymbol{c}\| \right) \right] \\
&\le (1 + \epsilon_2) \exp \left( \boldsymbol{t}^\top \mathbf{R}_2 \boldsymbol{c} \right) \quad (45)
\end{aligned}
$$

For some $\epsilon_2 > 0$. The last inequality holds because

$$
\begin{aligned}
\mathbb{E}_{c|c'} \left[ \exp \left( \kappa' \sqrt{d} \|\boldsymbol{c}' - \boldsymbol{c}\| \right) \right] &= \int \exp \left( \kappa' \sqrt{d} \|\boldsymbol{c}' - \boldsymbol{c}\| \right) p(c'|c) dc' \\
&= \underbrace{\exp(\kappa' \sqrt{d})}_{\ge 1} \underbrace{\int \exp(\|\boldsymbol{c} - \boldsymbol{c}'\|) p(c'|c) dc'}_{\ge 1} \\
&= 1 + \epsilon_2 \quad (46)
\end{aligned}
$$

To obtain a lower bound on $A(c)$ from the first-order Taylor approximation of $\exp(x) \ge 1 + x$ we observe that

$$
\mathbb{E}_{c|c'} \left[ \exp \left( \kappa' \sqrt{d} \|\boldsymbol{c}' - \boldsymbol{c}\| \right) \right] + \mathbb{E}_{c|c'} \left[ \exp \left( -\kappa' \sqrt{d} \|\boldsymbol{c}' - \boldsymbol{c}\| \right) \right] \ge 2. \quad (47)
$$

Therefore, from our model assumptions we have

$$
\mathbb{E}_{c|c'} \left[ \exp \left( -\kappa' \sqrt{d} \|\boldsymbol{c}' - \boldsymbol{c}\| \right) \right] \ge 1 - \epsilon_2 \quad (48)
$$

Hence,

$$
\begin{aligned}
A(c) &= \exp \left( \boldsymbol{t}^\top \mathbf{R}_2 \boldsymbol{c} \right) \mathbb{E}_{c'|c} \left[ \exp \left( \boldsymbol{t}^\top \mathbf{R}_2 (\boldsymbol{c}' - \boldsymbol{c}) \right) \right] \\
&\ge \exp \left( \boldsymbol{t}^\top \mathbf{R}_2 \boldsymbol{c} \right) \mathbb{E}_{c'|c} \left[ \exp \left( -\kappa' \sqrt{d} \|\boldsymbol{c}' - \boldsymbol{c}\| \right) \right] \\
&\ge (1 - \epsilon_2) \exp \left( \boldsymbol{t}^\top \mathbf{R}_2 \boldsymbol{c} \right) \quad (49)
\end{aligned}
$$

Therefore, from (46) and (49) we have

$$
A(c) = (1 \pm \epsilon_2) \exp \left( \boldsymbol{t}^\top \mathbf{R}_2 \boldsymbol{c} \right) \quad (50)
$$

Plugging $A(c)$ back in (43) we obtain

$$
\begin{aligned}
\log p(h, t \mid R) &= \log \left( \mathbb{E}_c \left[ \exp \left( \boldsymbol{h}^\top \mathbf{R}_1 \boldsymbol{c} \right) A(c) \pm \delta_0 \right] \right) - 2 \log Z + 2 \log(1 \pm \epsilon_z) \\
&= \log \left( \mathbb{E}_c \left[ \exp \left( \boldsymbol{h}^\top \mathbf{R}_1 \boldsymbol{c} \right) (1 \pm \epsilon_2) \exp \left( \boldsymbol{t}^\top \mathbf{R}_2 \boldsymbol{c} \right) \pm \delta_0 \right] \right) - 2 \log Z + 2 \log(1 \pm \epsilon_z) \\
&\qquad\qquad\qquad (51) \\
&= \log \left( \mathbb{E}_c \left[ \exp \left( \boldsymbol{h}^\top \mathbf{R}_1 \boldsymbol{c} \right) \exp \left( \boldsymbol{t}^\top \mathbf{R}_2 \boldsymbol{c} \right) \pm \delta_0 \right] \right) - 2 \log Z + 2 \log(1 \pm \epsilon_z) + \log(1 \pm \epsilon_2) \\
&= \log \left( \mathbb{E}_c \left[ \exp \left( \boldsymbol{h}^\top \mathbf{R}_1 \boldsymbol{c} + \boldsymbol{t}^\top \mathbf{R}_2 \boldsymbol{c} \right) \pm \delta_0 \right] \right) - 2 \log Z + 2 \log(1 \pm \epsilon_z) + \log(1 \pm \epsilon_2) \\
&= \log \left( \mathbb{E}_c \left[ \exp \left( \mathbf{R}_1{}^\top \boldsymbol{h} + \mathbf{R}_2{}^\top \boldsymbol{t} \right)^\top \boldsymbol{c} \pm \delta_0 \right] \right) - 2 \log Z + 2 \log(1 \pm \epsilon_z) + \log(1 \pm \epsilon_2)
\end{aligned}
$$

Note that $c$ has a uniform distribution over the unit sphere. In this case, from Lemma A.5 in (Arora et al., 2016b), (52) holds approximately.

$$\mathbb{E}_c \left[ \exp \left( \mathbf{R}_1^\top h + \mathbf{R}_2^\top t \right)^\top c \right] = (1 \pm \epsilon_3) \exp \left( \frac{\|\mathbf{R}_1^\top h + \mathbf{R}_2^\top t\|^2}{2d} \right) \tag{52}$$

where $\epsilon_3 = \tilde{O}(1/d)$. Plugging (52) in (51) we have that

$$\log p(h, t \mid R) = \frac{\|\mathbf{R}_1^\top h + \mathbf{R}_2^\top t\|_2^2}{2d} + O(\epsilon_z) + O(\epsilon_2) + O(\epsilon_3) + O(\delta_0') - 2\log Z \tag{53}$$

where $\delta_0' = \delta_0 \cdot \left( \mathbb{E}_c \left[ \exp \left( (\mathbf{R}_1^\top h + \mathbf{R}_2^\top t)^\top c \right) \right] \right)^{-1} = \exp(-\Omega(\log^{1.8} n))$. Therefore, $\delta_0'$ can be ignored. Note that $\epsilon_3 = \tilde{O}(1/d)$ and $\epsilon_z = \tilde{O}(1/\sqrt{n})$ by assumption. Therefore, we obtain that

$$\log p(h, t \mid R) = \frac{\|\mathbf{R}_1^\top h + \mathbf{R}_2^\top t\|_2^2}{2d} + O(\epsilon_z) + O(\epsilon_2) + \tilde{O}(1/d) - 2\log Z \tag{54}$$

$\square$

## B   LEARNING WITH MULTIPLE NEGATIVE TRIPLES

In this section, we show how the margin loss-based learning objective derived in section 3 can be extended to learn from more than one negative triples per each positive triple. This formulation leads to *rank-based* loss objective used in prior work on KGE. Considering that negative triples are generated via random perturbation, it is important to consider multiple negative triples during training to better estimate the classification boundary.

Let us consider that we are given a positive triple, $(h, R, t)$ and a set of $K$ negative triples $\{(h_k', R, t_k')\}_{k=1}^K$. We would like our model to assign a probability, $p(h, t \mid R)$, to the positive triple that is higher than that assigned to any of the negative triples. This requirement can be written as (55).

$$p(h, t | R) \geq \max_{k=1,\dots,K} p(h_k', t_k' \mid R) \tag{55}$$

We could further require the ratio between the probability of the positive triple and maximum probability over all negative triples to be greater than a threshold $\eta \geq 1$ to make the requirement of (55) to be tighter.

$$\frac{p(h, t \mid R)}{\max_{k=1,\dots,K} p(h_k', t_k' \mid R)} \geq \eta \tag{56}$$

By taking the logarithm of (56) we obtain

$$\log p(h, t \mid R) - \log \left( \max_{k=1,\dots,K} p(h_k', t_k' \mid R) \right) \geq \log(\eta) \tag{57}$$

Therefore, we can define the margin loss for a misclassification as follows:

$$L\left( (h, R, t), \{(h_k', R, t_k')\}_{k=1}^K \right) = \max \left( 0, \log \left( \max_{k=1,\dots,K} p(h_k', t_k' \mid R) \right) + \log(\eta) - \log p(h, t \mid R) \right) \tag{58}$$

However, from the monotonicity of the logarithm we have $\forall x_1, x_2 > 0$, if $\log(x_1) \geq \log(x_2)$ then $x_1 \geq x_2$. Therefore, the logarithm of the maximum can be replaced by the maximum of the logarithms in (58) as shown in (59).

$$L\left( (h, R, t), \{(h_k', R, t_k')\}_{k=1}^K \right) = \max \left( 0, \max_{k=1,\dots,K} \log \left( p(h_k', t_k' \mid R) \right) + \log(\eta) - \log p(h, t \mid R) \right) \tag{59}$$

By substituting (18) for the probabilities in (59) we obtain the rank-based loss given by (60).

$$L\left( (h, R, t), \{(h_k', R, t_k')\}_{k=1}^K \right) = \max \left( 0, 2d\log(\eta) + \max_{k=1,\dots,K} \|\mathbf{R}_1^\top h_k' + \mathbf{R}_2^\top t_k'\|_2^2 - \|\mathbf{R}_1^\top h + \mathbf{R}_2^\top t\|_2^2 \right) \tag{60}$$

In practice, we can use $p(h_k', t_k' \mid R)$ to select the negative triple with the highest probability for training with the positive triple.

Table 4: Statistics of the datasets

| Dataset | Relations | Entities | Train | Test | Validation |
|---------|-----------|----------|---------|--------|------------|
| FB15K | 1,345 | 14,951 | 483,142 | 59,071 | 50,000 |
| FB15K237 | 237 | 14,541 | 272,115 | 17,535 | 20,466 |
| WN18 | 18 | 40,943 | 141,442 | 5,000 | 5,000 |
| WN18RR | 11 | 40,943 | 86,835 | 3,134 | 3,034 |
| WN11 | 11 | 38,588 | 112,581 | 10,544 | 2,609 |
| FB13 | 13 | 75,043 | 316,232 | 23,733 | 5,908 |

## C  TRAINING DETAILS

The statistics of the benchmark datasets are show in Table 4.

We selected the initial learning rate ($\alpha$) for SGD in $\{0.01, 0.001\}$, the regularisation coefficients $(\lambda_1, \lambda_2)$ for the orthogonality constraints of relation matrices in $\{0, 1, 10, 100\}$. The number of randomly generated negative triples $n_{\mathrm{neg}}$ for each positive example is varied in $\{1, 10, 20, 50, 100\}$ and $d \in \{50, 100\}$. Optimal hyperparameter settings were: $\lambda_1 = \lambda_2 = 10$, $n_{\mathrm{neg}} = 100$ for all the datasets, $\alpha = 0.001$ for FB15K, FB15K237 and FB13, $\alpha = 0.01$ for WN18, WN18RR and WN11. For FB15K237 and WN18RR $d = 100$ was the best, whereas for all other datasets $d = 50$ performed best.

