# OpenReview forum: "RelWalk -- A Latent Variable Model Approach to Knowledge Graph Embedding"
_ICLR.cc/2019/Conference_

### Official Review · AnonReviewer1 · 2018-11-02
**The paper "RELWALK – A LATENT VARIABLE MODEL APPROACH TO KNOWLEDGE GRAPH EMBEDDING" is about a generative knowledge graph embedding process and a theoretic motivation for a corresponding scoring function.**

**Rating:** 4
**Confidence:** 4

**Review:**

+ Theoretic explanation for the scoring function.
+ (Promise for) Online provided source code.
+ The paper is well-written.

- The authors missed [1] which also introduces a generative model for knowledge graph embeddings.
- The use of the datasets FB15k-237 and WN18RR instead of FB15k and WN18 (without inverse relations) would enable a better empirical evaluation. By using the flawed FB15k and WN18 datasets, the evaluation is biased towards the usage of inverse relations which should not exist in a link prediction evaluation dataset.
- The authors are not mentioning and comparing to walk based approaches like node2vec [2], Deepwalk [3], and rdf2vec [4].


Due to the missing comparisons to the mentioned references above and the possible bias in the evaluation, I am leaning towards rejecting the paper.


Minor comments:

. Abbreviations like h for head are used before they are introduced.
. "The the" -> "The"
. "triples are likely too be obvious examples" -> "triples are likely to be obvious examples"


[1] Xiao, Han, Minlie Huang, and Xiaoyan Zhu. "TransG: A generative model for knowledge graph embedding." Proceedings of the 54th Annual Meeting of the Association for Computational Linguistics (Volume 1: Long Papers). Vol. 1. 2016.

[2] Grover, Aditya, and Jure Leskovec. "node2vec: Scalable feature learning for networks." Proceedings of the 22nd ACM SIGKDD international conference on Knowledge discovery and data mining. ACM, 2016.

[3]  Perozzi, Bryan, Rami Al-Rfou, and Steven Skiena. "Deepwalk: Online learning of social representations." Proceedings of the 20th ACM SIGKDD international conference on Knowledge discovery and data mining. ACM, 2014.

[4] Petar Ristoski, Jessica Rosati, Tommaso Di Noia, Renato De Leone, and Heiko Paulheim. "RDF2Vec: RDF Graph Embeddings and Their Applications." SWJ http://www.semantic-web-journal.net/content/rdf2vec-rdf-graph-embeddings-and-their-applications-1

---

> ### Author Response · Authors · 2018-11-13
> **Author response: AnonReviewer1**
>
> Q1: - The authors missed [1] which also introduces a generative model for knowledge graph embeddings.
>
> Ans: TransG [1] is described in detail in the related work section and is compared against in the experiments section. The proposed method, RelWalk, out performs TransG in FB13 dataset in the triple classification task.
>
> Q2: - The authors are not mentioning and comparing to walk based approaches like node2vec [2], Deepwalk [3], and rdf2vec [4]. Due to the missing comparisons to the mentioned references above and the possible bias in the evaluation, I am leaning towards rejecting the paper.
>
> Ans: Please note that all these papers are cited in the revised version and their relevance discussed in the related work section. However, we note that [2], [3], [4] are only weakly related to the KGE task we consider in this paper because those paper are either assuming a richer knowledge graph (RDF annotated) than a simple relation-labelled knowledge base or learning vertex representations only. Therefore, following the prior work on KGE, in our evaluations, we compare methods that use the standard benchmarks such as Freebase and WordNet.
>
> All minor comments are updated in the paper and the papers are cited.

---

### Official Review · AnonReviewer3 · 2018-11-02
**An OK paper on theoretical understanding the KGE**

**Rating:** 5
**Confidence:** 5

**Review:**

In this paper, the author proposed a way to understand the knowledge graph embedding task. More specifically, the authors try to extend the random walk model (Arora et al., 2016a) of word embeddings to KGE.

Some of my detailed comments and questions below.

1. To me, this paper sounds like a direct application of the method of Arora et al., 2016a to KGE. Therefore, this does look like a very obvious application, and it is not clear to me if this paper presents any new methods or if you have obtained any new insights.

2. The paper claims that all prior work use some sort of heuristics in the KGE task, and their approach is a generative account that might deal with this issue? But I personally found that by using random walk and the exp function, you are also making some very strong assumptions that are similar to heuristics? How do you know it is exp function but not other function?

3. I am not sure what the purpose for the evaluation section is. You mention that this paper is not about state-of-the-art results, but if you theory really works, your scoring function should beat SOTA results.

Overall, I have to say that I am very disappointed with this paper, because there are no new theoretical tools being introduced, and the authors seem to be applying Arora et al., 2016a from word embedding to KGE only.

---

> ### Author Response · Authors · 2018-11-13
> **Author response: AnonReviewer3**
>
> Q1:  To me, this paper sounds like a direct application of the method of Arora et al., 2016a to KGE. Therefore, this does look like a very obvious application, and it is not clear to me if this paper presents any new methods or if you have obtained any new insights.
>
> Ans: We would like to thank the reviewer for the time and review.
>
> It is true that we are extending the original Random Walk model proposed by Arora et al [2016a] to model relations. However, we do not agree that is a “very obvious application”. Next, we detail the reasons for this.
>
> The original random walk model was proposed for a co-occurrence graph, ignoring the relations between the entities. It is nontrivial as how to incorporate relational information into this model. We have both proposed a new relational version of the original random walk model and have proven that it can obtain state-of-the-art knowledge graph embeddings.
>
> Q2: The paper claims that all prior work use some sort of heuristics in the KGE task, and their approach is a generative account that might deal with this issue? But I personally found that by using a random walk and the exp function, you are also making some very strong assumptions that are similar to heuristics? How do you know it is exp function but not other function?
>
> Ans: This is a very standard way to represent the potential function in a probabilistic graph. Moreover, as you can see from the Taylor expansion of exp, it contains all polynomials terms in it. In fact, prior work on kernel methods have shown that exponential kernels such as Radial Basis Functions (RBF) kernel can theoretically subsume all other types of kernel functions. Therefore, this is not a heuristic but a very general modelling assumption that does not assume anything about the properties of entities and their relations.
>
>  Q3:  I am not sure what the purpose for the evaluation section is. You mention that this paper is not about state-of-the-art results, but if your theory really works, your scoring function should beat SOTA results. Overall, I have to say that I am very disappointed with this paper, because there are no new theoretical tools being introduced, and the authors seem to be applying Arora et al., 2016a from word embedding to KGE only.
>
> Ans: At the initial submission of the paper, we did not have state-of-the-art results for KGE. Therefore, we did not claim this in the initial submission. However, we have obtained state-of-the-art results for FB15k237 and WN18RR datasets as shown in the updated version of the paper. Note that FB15k237 and WN18RR datasets were recently proposed by removing reverse relations that were easier to predict in the original versions of those two datasets. We have revised the paper with these results and have claimed as such. Therefore, this paper not only provides a theoretical analysis but also obtains state-of-the-art results on two modern benchmarks for KGE.

---

### Official Review · AnonReviewer2 · 2018-11-13
**A solid idea that seems to work in practic but the novelty and the empirical justification may not be enough.**

**Rating:** 6
**Confidence:** 4

**Review:**

This paper proposes to perform the link prediction in knowledge bases by introducing a new scoring function and theoretically motivating their method. The authors validate their proposed approach through several experiments.

This paper reads well and the results appear sound. I personally find the theoretical argument behind the proposed scoring function very interesting. Unfortunately, the contribution seems rather small to be accepted for ICLR. This is a straight application and combination of existing pieces with not much originality and without being backed up by very strong experimental results. My concerns are as follows:

   - Having only results on two flawed datasets (considering the inverse relations in them) makes it hard to evaluate the quality of the method. I suggest conducting the experiments on the FB15K-237 and WN18RR from [1] instead.
   - You only evaluate on MR and Hits@10, but it is standard to include metrics like MRR and Hits@1 and 3 also, since no metric is perfect for this task.
   - Since the goal of the work is not providing a state of the art method, and focus on the theoretical understanding of their scoring function, it is of high importance to assess the characteristics of their embeddings and scoring function through designing other experiments. As a result, I suggest to study the geometric behavior of their embeddings and compare it to the other methods. Further, investigating the semantic purity of the embeddings by calculating the entropy of the type distribution of the entities, similar as [2], can shed more light on the significance of their method.

On overall, although the proposed method seems a direct application of Arora et al.,2016a, I find their extension novel and quite interesting, But the paper needs more experimental results to validate the idea.


[1] Dettmers, Tim, et al. "Convolutional 2d knowledge graph embeddings.", AAAI-18.
[2] Ding, Boyang, et al. "Improving Knowledge Graph Embedding Using Simple Constraints.", ACL-18.

---

> ### Author Response · Authors · 2018-11-13
> **Author response: AnnonReviewer2**
>
> Q1: - Having only results on two flawed datasets (considering the inverse relations in them) makes it hard to evaluate the quality of the method. I suggest conducting the experiments on the FB15K-237 and WN18RR from [1] instead.
>
> Ans: In the revised version we have conducted experiments using both FB15K-23k and WN18RR datasets and the proposed method (RelWalk) obtains the state-of-the-art performance on both datasets.
>
> Q2: - You only evaluate on MR and Hits@10, but it is standard to include metrics like MRR and Hits@1 and 3 also, since no metric is perfect for this task.
>
> Ans: Given the limited availability of space, we had to select the evaluation measures that are more widely used for this task, which are MR and Hits@10.
>
> Q3:  Since the goal of the work is not providing a state of the art method, and focus on the theoretical understanding of their scoring function, it is of high importance to assess the characteristics of their embeddings and scoring function through designing other experiments. As a result, I suggest to study the geometric behavior of their embeddings and compare it to the other methods. Further, investigating the semantic purity of the embeddings by calculating the entropy of the type distribution of the entities, similar as [2], can shed more light on the significance of their method.
>
> Ans: Thank you for the suggestion. As shown in the revised version, the proposed method is obtaining state-of-the-art results in addition to its theoretical contribution. Therefore, we would believe this is sufficient for an 8-page conference publication and would consider the suggested additional evaluations in a longer journal version.

---

> > ### Comment · AnonReviewer2 · 2018-11-18
> > **Interesting Results**
> >
> > It is impressive that you could achieve state of the art results with your proposed scoring function, but I suggest updating your Table 2 with more recent baselines, specifically adding ConvE [1]. Furthermore, I believe the number of page restriction does not strongly being enforced in ICLR venue, and you could even add the results of your method for more metrics to the Appendix; I am really interested in seeing those results.
> >
> > [1] Dettmers, Tim, et al. "Convolutional 2d knowledge graph embeddings.", AAAI-18.

---

> > > ### Author Response · Authors · 2018-11-25
> > > **Re: Interesting Results**
> > >
> > > Thank you for your comment. We have added ConvE results to Table 3 in the paper. Interestingly, the proposed method (RelWalk) outperforms ConvE also on both FB15k237 and WN18RR datasets.

---

### Public Comment · (anonymous) · 2018-09-29
**Not mention recent results**

You missed to mention a lot of recent results which are much better than yours from last two years, as you can see a part from [1].

[1] An overview of embedding models of entities and relationships for knowledge base completion.

---

> ### Author Response · Authors · 2018-09-29
> **Re: not mention recent results**
>
> Thank you for your comment. As we have stated in several places in the paper, the focus of this work is to provide a theoretical explanation to the knowledge graph embedding and not to propose yet another heuristically-motivated scoring formula. We have discussed the most recent work by Ding et al. (ACL 2018, which happened in July 2018), which is the most recent venue. It is true that there are over 35 scoring formulas proposed in the literature as shown in [1]. However, we did not find any method to be providing a rigorous theoretical analysis as done in our work. In the empirical validation section, we have selected methods that have been repeatedly used in prior work as comparison points. As we have stated in the paper, and shown in the empirical validation section, although the derived scoring formula obtains good results they are not SoTA. However, the main contribution of the paper is in the theoretical extension of the generative model proposed by Arora et al. (2016), and to this extent, we have covered all theoretically-relevant prior work on such extensions.

---

> > ### Public Comment · (anonymous) · 2018-09-30
> > **Re: not mention recent results**
> >
> > You had one sentence about Ding et al. (2018) in related work section and ignored most works in 2017 and 2018. You did not motivate well how we should need the generative process of a knowledge graph. You can see [1] that you did not cite.
> >
> > More importantly, you did an important heuristic assumption: relations are asymmetric in general, this may be not true in real-world datasets. For example, you did experiments on WN18 (WN11) and FB15 (FB11) consisting of many reversible relations. That's reason why you were not among SOTA results (results of ComplexE and NTN as you mentioned were not SOTA since 2017). I would suggest you do experiments on WN18RR and FB15k-237 as mentioned in [2].
> >
> > [1] TransG : A Generative Model for Knowledge Graph Embedding. ACL 2016.
> > [2] Convolutional 2D Knowledge Graph Embeddings. AAAI 2018.

---

> > > ### Author Response · Authors · 2018-09-30
> > > **Re Re: not mention recent results**
> > >
> > > Thank you again for the comments. Much appreciated. We have added results on WN18RR, which shows that the proposed method (RelWalk) is doing well on this dataset as well. We have also added TransG to the evaluations on Link Prediction and discussed briefly in the related work section.
> > >
> > > see the version here:  https://www.dropbox.com/s/x3kixodqays7nwi/RelWalk-ICLR.pdf?dl=0
> > >
> > > We will reflect these additional details during the rebuttal stage.
> > >
> > > However, as stated in the previous response and also in the paper, the main contributions in the paper is the theoretical extension of the word embedding model of to KGE.
> > >
> > > "More importantly, you did an important heuristic assumption: "...
> > > No we DO NOT make this assumption at all. The third sentence in Sec 3 explicitly states that we assume relation to be asymmetric in general. Asymmetry is a general assumption and the model can capture symmetry as a special case. You can see this from the scoring function we derive as well. Unless otherwise, R_1 and R_2 are equal  (for the symmetric case) p(h,t | R) and p(t,h | R) will not be equal.

---

### Public Comment · (anonymous) · 2018-10-04
**Spherical Gaussian distribution**

Hello,

Thanks for the nice paper.
Could you provide the definition of spherical Gaussian distribution? I thought that it is a distribution on l_2 ball since the random variable has a unit norm, but the description saying it has zero means and diagonal covariance matrix makes me confused. since it is somewhat different from the usual description here: https://mynameismjp.wordpress.com/2016/10/09/sg-series-part-2-spherical-gaussians-101/

Or is that the multivariate Gaussian where each dimension is independent to each other while having the same variance? if then, did you normalize the random vector to make a unit vector? In this case, is it still normally distributed? it should be uniformly distributed on the unit ball.

Thanks in advance!

---

> ### Author Response · Authors · 2018-10-06
> **Re: Spherical Gaussian distribution**
>
> Thank you for your question.
>
> It is the latter. Specifically, we consider a multivariate Gaussian with Identity covariance. A spherical distribution, in general, is a one with equal variances in each dimension and without any cross-correlations. Hope this clarifies your concern.

---

> > ### Public Comment · (anonymous) · 2018-10-07
> > **Re: Spherical Gaussian distribution**
> >
> > In that case the ||\hat{h} ||_1 should not a unit vector (described under equation 7). The support should be R^D. Right? I guess you have confused C and \hat{h} where the former is distributed on a unit sphere whereas the latter is not. Now I understand it's just a typo, but it makes me confused while reading the proof of lemma 1. Thanks.

---

> > > ### Author Response · Authors · 2018-10-08
> > > **Clarification**
> > >
> > > Thank you for spotting the typo.  $\hat{h}$ is in the Spherical Gaussian with unit covariance matrix $\mat{I} \in \R^{d \times d}$ and not $s^2\mat{I} \in \R^{d \times d}$.
> > >
> > > > I guess you have confused C and \hat{h} where the former is distributed on a unit sphere whereas the latter is not.
> > >
> > > No. They are both on the unit sphere. c is on the unit sphere from the assumption, whereas h is represented in the polar coordinate form where the direction is given by the unit vector \hat{h} and the scale is given by s_h. This transformation can be done for any vector h.

---

> > > > ### Public Comment · (anonymous) · 2018-10-10
> > > > **Clarification**
> > > >
> > > > I think you still don't understand the question.
> > > >
> > > > If, as you said, \hat{h} is distributed as the multivariate Gaussian with zero mean and unit covariance matrix, ||\hat{h}||_2 is not guaranteed to be 1 (because \hat{h} \in R^D), whereas in the paper \hat{h} is a unit vector.
> > > >
> > > > The unit covariance matrix does not make the multivariate Gaussian distributed on a unit sphere.
> > > > In the other words, \hat{h} = (1, 1, 1) can be drawn from MN(0, I) where ||(1,1,1)||_2 = \sqrt{3} != 1.

---

> > > > > ### Author Response · Authors · 2018-10-12
> > > > > **Re: Spherical Gaussian**
> > > > >
> > > > > >If, as you said, \hat{h} is distributed as the multivariate Gaussian with zero mean and unit covariance matrix, ||\hat{h}||_2 is not guaranteed to be 1 (because \hat{h} \in R^D), whereas in the paper \hat{h} is a unit vector.
> > > > >
> > > > > True. But you could always scale the sampled vector such that it has unit length.

---

### Meta-Review · Area_Chair1 · 2018-12-17
**Limited novelty, and further analysis needed**

**Confidence:** 2
**Recommendation:** Reject

**Metareview:**

This paper proposes a new scoring function for link prediction model that is based on a generative model for the knowledge graph, based on a random-walk model previously used for word embeddings. The new scoring function, as it is accompanied by the generative model, provides interesting theoretical results that the reviewers also appreciate. Finally, the results are quite strong, as they obtain state-of-art on the primary benchmarks for the task.

Based on the submitted version, the reviewers and AC note the following potential weaknesses: (1) the reviewers felt that the proposed work is a direct application of the random-walk model from Arora et al. and thus limited in novelty, (2) given the generative model, the reviewers felt that the paper would benefit from an analysis of the learned embeddings, and their difference from ones from existing approaches, (3) The reviewers noted that the authors were using an incorrect version of FB15k and WN18, (4) the authors were not providing results for all the metrics, (5) the coverage of related work is quite limited.

The authors addressed many of the concerns raised by the reviewers in their comments and revision, in particular, they obtained state-of-art results for the corrected versions of the benchmarks. Further, they clarified the assumptions made in their modeling and revised the related work to include the papers that the reviewers mentioned. However, the concerns regarding the lack of novelty of the proposed approach, w.r.t Arora et al 2016 and the need for further analysis of the learned embeddings, still remain.

This paper comes really close to getting accepted, but ultimately the reviewers agree that the remaining concerns need to be addressed.